# Genome-Wide Identification and Expression Analysis of the Cytochrome P450 Gene Family in *Bemisia tabaci* MED and Their Roles in the Insecticide Resistance

**DOI:** 10.3390/ijms24065899

**Published:** 2023-03-21

**Authors:** Penghao Qin, Haoyuan Zheng, Yunli Tao, Youjun Zhang, Dong Chu

**Affiliations:** 1Shandong Engineering Research Center for Environment-Friendly Agricultural Pest Management, College of Plant Health and Medicine, Qingdao Agricultural University, Qingdao 266109, China; 2Department of Plant Protection, Institute of Vegetables and Flowers, Chinese Academy of Agricultural Sciences, Beijing 100081, China

**Keywords:** *Bemisia tabaci* MED, genomics, cytochrome P450, insecticide resistance, RNA interference

## Abstract

The whitefly, *Bemisia tabaci* MED (Hemiptera: Aleyrodidae), is an omnivorous agricultural pest, which causes huge economic losses to agriculture and is highly resistant to many pesticides. The overexpression of cytochrome P450 may play an important role in host adaptation and insecticide resistance in *B. tabaci* MED. Therefore, the present study systematically analyzed the cytochrome P450 gene family at the genome-wide level to understand its function in *B. tabaci* MED. Our analysis identified 58 cytochrome P450 genes in *B. tabaci* MED, among which 24 were novel. Phylogenetic analysis revealed broad functional and species-specific diversification in *B. tabaci* MED P450, suggesting the role of multiple P450 genes in detoxifying. Reverse transcription-real time quantitative PCR (RT-qPCR) showed that CYP4CS2, CYP4CS5, CYP4CS6, CYP4CS8, CYP6DW4, CYP6DW5, CYP6DW6, CYP6DZ8, and CYP6EN1 genes increased significantly after two days of exposure to imidacloprid. Interestingly, all nine genes belonged to the CYP4 and CYP6 families. A decrease in the expression of five genes (CYP6DW4, CYP6DW5, CYP6DW6, CYP6DZ8, and CYP4CS6) via RNA interference (RNAi) resulted in a significant increase in the mortalities of whiteflies when exposed to imidacloprid. These results indicate that the overexpression of the P450 genes may play an essential role in imidacloprid tolerance of *B. tabaci* MED. Thus, the present study provides basic information on P450 genes in *B. tabaci* MED, which will further help elucidate the insecticide resistance mechanism in the agricultural pest whitefly.

## 1. Significance

The whitefly, *Bemisia tabaci* (Gennadius) (Hemiptera: Aleyrodidae), is the main pest worldwide that is resistant to many insecticides. Therefore, it is necessary to develop new and effective compounds against *B. tabaci*. We sequenced the whole genome of *B. tabaci* to detect cytochrome P450 protein, and knocked out five genes by RNA interference, which led to an increase in mortality after imidacloprid exposure, and confirmed their role in insecticide resistance. These findings provide a theoretical basis for further studying the specific function of cytochrome P450 protein and clarifying the protein–compound interaction to develop effective pesticides.

## 2. Introduction

The whitefly, *B. tabaci*, is one of the most destructive polyphagous phloem-feeders worldwide [1]. *B. tabaci* is a cryptic species complex including the species “MEAM1” and “MED” (also known as biotype B and biotype Q, respectively), acquiring a globally invasive insect status and are widely distributed throughout China, Southern United States, and Africa [2,3]. *B. tabaci* do direct damage by feeding on plant sap and indirect damage by transmitting plant viruses to vegetable crops [4,5,6]. Because of its ability to transmit begomoviruses and criniviruses very efficiently into many vegetable crops [7,8,9,10], farmers respond with calendar application of insecticides, often with insecticides sharing the same mode of action, resulting in resistance development in exposed populations [11,12].

Cytochrome P450 contributes to growth, development, nutrition, and detoxification of xenobiotics, and is used to metabolize a variety of endogenous and exogenous compounds, forming a polygenic enzyme superfamily [13,14]. Cytochrome P450 is the main detoxification enzyme system involved in the metabolism of various pesticides and other exogenous and endogenous compounds in insects, and participates in the synthesis and metabolism of various endogenous substances, such as ecdysone, juvenile hormone, and fatty acids [15]. In addition, P450 can enhance the detoxification activity by overexpressing and accumulating detoxification proteins or changing these proteins’ structure [16,17]. In the P450 supergene family, CYP4 and CYP6 family genes are mainly related to pests’ insecticide resistance [18]. Studies have also proven that P450 plays a key role in insecticide resistance due to its genetic diversity, broad substrate specificity, and catalytic versatility [19]. However, no systematic study has been conducted on the cytochrome P450 gene family at the genome-wide level in *B. tabaci* MED.

As an alternative to organophosphates and pyrethroids, neonicotinoid insecticides have been used to control whiteflies in fields and greenhouses; however, the extensive use has led to the rapid development of neonicotinoid resistance [20,21]. *B. tabaci* MED has a high level of resistance to neonicotinoid insecticides, which may contribute to genetic cluster changes [22].

Neonicotinoid insecticides were introduced in 1991, with imidacloprid being the first product in the market. Imidacloprid targeted nicotinic acetylcholine receptor (nAChR) receptors in the insect nervous system [23]. Because of its broad-spectrum action, high efficiency, low toxicity, and low residue, it was widely adopted in many cultivations throughout the world, including China. Furthermore, imidacloprid had multiple modes of action, such as contact killing, stomach toxicity, and systemic absorption, resulting in the slow evolution of resistance in some cropping systems [24]. However, because of its genetic ability to better metabolize insecticides and selection pressure imposed by the calendar application of imidacloprid, populations of *B. tabaci* MED have become extremely tolerant to the commercial products of this insecticide [25]. Studies have demonstrated that the overexpression of CYP6CM1 in *B. tabaci* leads to imidacloprid resistance [26]. However, only a few P450s have been published in *B. tabaci* compared with other insects, such as *Bombyx mori* (84) [27], *Leptinotarsa decemlineata* (96) [18], *Rhynchophorus ferrugineus* (77) [28], and *Drosophila melanogaster* (85) [29]. Therefore, to better understand the resistance mechanism in *B. tabaci* MED, there is a need to identify more P450 genes of *B. tabaci* MED.

The present study aimed to identify the cytochrome P450 gene family in *B. tabaci* MED and assess their roles in insecticide resistance. We conducted a genome-wide analysis of P450 sequences in the complete genome of *B. tabaci* MED and investigated their roles in insecticide resistance using expression analysis, RNAi approach, and a bioassay.

## 3. Results

### 3.1. Genome-Wide Identification of the P450 Gene Superfamily in B. tabaci

We obtained about 105.92 Gb of clean data by sequencing the *B. tabaci* library on the PacBio platform. The total sequencing depth was about 166.16 x, the read N50 was 23.42 Kb, and the average read length was 14.74 Kb (Table 1). After searching by BlastP and BlastN, 24 genes encoding P450 protein were identified in the MED genome of *B. tabaci* MED. The predicted proteins encoded by the 24 *BtP450* genes were initially classified using NCBI CDD analysis into four families, including CYP4, CYP6, CYP9, CYP301-318, and CYP18a (Table 2).

Further, in order to evaluate the phylogenetic relationship between each family of cytochrome P450 proteins in *B. tabaci* MED, all P450 proteins were compared with MEGA7 to generate a phylogenetic tree. As shown in Figure 1, the P450 proteins from the same family clustered together. The topology of the phylogenetic tree reconstructed by the ML method and PhyML was roughly in agreement with the NJ method, demonstrating our results’ reliability. The CYP4 and CYP6 families of *B. tabaci* MED were the largest families (Figure 1).

### 3.2. Phylogenetic and Protein Structure Analyses of P450s in B. tabaci MED

Then, in order to explore the structural diversity of *B. tabaci* MED P450 superfamily, MEME was used to analyze the conserved motifs. As shown in Figure 2, the motifs of P450 proteins in the same family are also extremely similar, and these motifs are highly conserved among *B. tabaci* MED P450 proteins. Meanwhile, a few P450 proteins from sister branches also shared a common motif composition (Figure 2). This phenomenon is probably associated with the gene structure and phylogenetic relationships.

### 3.3. Differential Expression of BtP450 Genes under Imidacloprid Treatment

After *B. tabaci* MED adults were fed 50 ppm imidacloprid, we quantified the P450 protein-encoding 24 non-redundant genes (Figure 3). Among these 24 genes, CYP4CS2, CYP4CS5, CYP4CS6, CYP4CS8, CYP6DW4, CYP6DW5, CYP6DW6, CYP6DZ8, and CYP6EN1 were found to be significantly upregulated. Then, to examine the involvement of BtaP450 in the whitefly imidacloprid stress response, the upregulated and downregulated genes were selected as candidates for RNA interference. Further qPCR analysis showed that the expression of significantly upregulated genes in *B. tabaci* MED was suppressed considerably after two days of dsRNA feeding (Figure 3).

### 3.4. Effect of BtP450 RNAi on Imidacloprid Tolerance of B. tabaci MED

To further investigate the role of BtP450 in the *B. tabaci* MED insecticide stress response, five genes (CYP6DW4, CYP6DW5, CYP6DW6, CYP6DZ8, and CYP4CS6) significantly upregulated after imidacloprid treatment were selected as candidates for RNAi. The expression of these five genes of the CYP4 and CYP6 family was suppressed considerably after two days of dsRNA feeding, as shown by qPCR (Figure 4).

After feeding dsRNA for two days, we performed biological assays with 100 ppm of imidacloprid (Figure 5). RNAi of CYP6DW4, CYP6DW5, CYP6DW6, CYP6DZ8, and CYP4CS6 significantly increased the susceptibility of *B. tabaci* MED to imidacloprid. After knocking out these genes, the mortality rate for whiteflies exposed to dsRNA was significantly higher in comparison to whiteflies exposed to dsEGFP.

## 4. Discussion

In 2003, *B. tabaci* MED was identified as an alien species in China, which seriously impacted agroecology [30]. With the rapid development of genome sequencing technology, whole genome sequencing helped researchers study the gene families. The genome sequencing of *B. tabaci* MED has also been completed, which provides a good foundation for understanding the evolutionary relationship of each gene family of *B. tabaci* MED. This study used the transcriptome data of *B. tabaci* MED to identify the P450 gene family and analyze the location and evolutionary relationship of the various P450 genes in the genome. The results detected 24 P450 gene sequences in the *B. tabaci* MED genome, which were divided into four clans, including Clan 2, Clan 3, Clan 4, and Clan M. Compared with the published P450 genes, the P450 genes of Clan 3 and Clan 4 in the *B. tabaci* genome showed an apparent gene expansion. Previous studies have shown that the P450 gene families: CYP 4, 6, and 9 are associated with pesticide resistance and host adaptation. For example, in *B. tabaci*, the CYP6CM1 gene is related to imidacloprid resistance [26]. In the brown planthopper, the CYP6ER1 gene is associated with imidacloprid resistance [31]. Similarly, the bumblebee CYP9Q3 efficiently metabolizes thiacloprid [32,33], and CYP9Q6 metabolizes thiacloprid and acetamiprid [34]. The P450 CYP4, 6, and 9 families are Clan 3 and Clan 4 genes in *B. tabaci* MED; therefore, it is very likely that these clans of genes might be related to the insecticide resistance in *B. tabaci* MED as well.

Glutathione S-transferase, esterase, and P450 are generally regarded as detoxification enzymes and help ward off and catabolize toxins, including pesticides. Studies have demonstrated point mutations in pesticide target genes can lead to an increase in gene expression and diversification of coding sequences with the variable outcome for insecticide metabolism in insects [35]. For instance, in *B. tabaci* MED, the increase in susceptibility has been primarily associated with the mutations in the P450 monooxygenase rather than the nicotinic acetylcholine receptor, which is the target site of imidacloprid action [36]. The P450-specific inhibitor piperonyl butanol significantly reduced imidacloprid resistance levels, demonstrating the important role of P450s in insecticide resistance [37]. Similarly, P450-mediated effects of insecticide resistance have been reported in aphids [38]. In addition, specific P450 genes have been associated with neonicotinoid resistance. For example, Drosophila CYP6G1 confers imidacloprid resistance [39,40]. Puinean proposed that CYP6Y3 confers neonicotinoid resistance to *Myzus persicae* [41]. Puinean significantly reduced imidacloprid resistance levels using the P450s-specific inhibitor piperonyl butanol, which demonstrated an important role for P450s in insecticide resistance [37]. It is reported that members of CYP3 and CYP4 families participate in the metabolism of pesticides because of long-term use of pesticides [19]. Elevated expression levels of CYP6CM1 and CYP4C64 were associated with imidacloprid resistance in *B. tabaci* [25,26], whereas overexpression of CYP6ER1 and CYP6AY1 was closely associated with imidacloprid resistance in *N. lugens* [42]. In red palm weevil (RPW), CYP345J1 and CYP6NR1 overexpression metabolized pesticide molecules more efficiently [28], thereby enhancing tolerance of RPW in palm fields, both CYP345 and CYP6 belong to a single gene clan of P450.

This study identified nine highly expressed P450 genes after imidacloprid treatment in MED. Subsequent follow-up studies using RNAi and qPCR revealed that feeding MED with dsRNA corresponding to these genes resulted in a significant reduction in the expression of five genes (belonging to CYP4 and CYP6 families). Furthermore, the known down of these genes resulted in a significant increase in mortality of *B. tabaci MED* exposed to imidacloprid signifying these genes had a direct role in imidacloprid metabolism in *B. tabaci* MED. Similarly, previous studies with other coleopterans, such as *Tribolium castaneum* and *Leptinotarsa decemlineata,* have also reported that the role of CYP4 173 and CYP6 family genes in the metabolism of xenobiotics compounds [18,43,44]. Taken together, data from this study and previous studies suggest that the xenobiotic metabolism role of these genes might be conserved in insects. However, further studies are warranted to fully comprehend the role these genes might play in insecticide metabolism and insecticide resistance development in insect pests.

The current study provides a genomic database of *B. tabaci* MED P450 and a comprehensive picture of the *B. tabaci* MED P450 family based on gene identification and imidacloprid-induced expression profiling. We systematically analyzed 24 *B. tabaci* MED P450 genes, which provide a basis for further molecular and functional characterization of P450. The phylogenetic tree divided the *B. tabaci* MED P450 into four clans, of which clan 2 is dominant. Our research provides a solid foundation for future insecticide design. However, to guide the rational development of pesticide-related compounds and improve the accuracy and efficiency of computational-based drug construction, a deeper understanding of the interaction of imidacloprid-related heterocyclic compounds with P450s is required. Our study reveals various roles of *B. tabaci* MED P450 and provides a reference for better research on the functions of *B. tabaci* MED P450.

## 5. Materials and Methods

### 5.1. Insect Sampling and Rearing

The *B. tabaci* MED Lingshui population was collected from Lingshui City, Hainan Province, in January 2017. Since the collection, all populations have not been exposed to pesticides, and they are raised under laboratory conditions, without any further selection pressure, and each generation is about every 25 days. These insects were reared on the common tobacco variety NC89 for a long time in insect-proof cages (temperature 27 ± 1 °C, relative humidity 60 ± 5%, and light cycle 16L:8D).

### 5.2. P450 Gene Classification and Phylogenetic Analysis

The experimental process is carried out according to the standard protocol provided by Pacbio, and the library construction includes six steps: (1) use g-TUBE to interrupt the DNA sample; (2) repair the damaged DNA sample; (3) terminal repair of DNA; (4) connect the dumbbell-shaped connector; (5) exonuclease digestion of nucleic acid; and (6) screening the target fragment with BluePippin to obtain the sequencing library. The present study used three strategies to predict the gene structure: de novo prediction, homologous species-based prediction, and Unigene-based prediction. De novo prediction was carried out using Genscan [45], Augustus (v2.4) [46], GlimmerHMM (v3.0.4) [47], GeneID (v1.4), and SNAP (v2006-07-28) [48] programs. The homologous species-based prediction was obtained by running GeMoMa (v1.3.1) [49,50], Hisat (v2.0.4) [51]. and Stringtie (v1.2.3) [52] for reference transcript-based assembly, followed by gene prediction using TransDecoder (v2.0) and GeneMarkS -T (v5.1) [53]. Meanwhile, PASA (v2.0.2) [54] was used to predict Unigene sequences based on transcriptome data without reference assembly. Finally, EVM (v1.1.1) was used to integrate the prediction results obtained by the above three methods and modified with PASA. The predicted gene sequences were compared with NR [55], KOG [56], GO [57], KEGG [58], TrEMBL [59], and other functional databases by BLAST (v2.2.31) [60] with an e-value of 1e-5. Then, the KEGG pathway annotation analysis and KOG function annotation analysis of genes were carried out. The *B. tabaci* samples were obtained from Lingshui, Hainan, in 2017 and identified as MED using the mtCOI gene [61,62]. Genes annotated as cytochrome P450 based on the genome annotation library were selected as candidate genes.

In order to analyze the phylogenetic tree, P450 amino acid sequences of various insects were downloaded from NCBI, and then merged with 24 P450 amino acid sequences of *B. tabaci*. The phylogenetic tree was analyzed by MEGA7 NJ method and corrected for 1000 times. Then the conserved motifs of 24 P450 sequences were analyzed by MEME, and the screened P450 genes were classified according to the phylogenetic tree, and further named according to the molecular weight, and then handed over to Dr. David Nielsen (CYP Nomenclature Committee) for gene naming.

### 5.3. Imidacloprid Treatment

Then, to express P450 under imidacloprid stress, whiteflies were treated with imidacloprid at a concentration of 50 ppm for two days, using a blank control and triplicates per treatment; approximately 100 adults were maintained per replicate.

### 5.4. RNA Extraction, cDNA Synthesis, and P450 Cloning

The RNA samples were extracted from 100 adults using TRIzol reagent (Thermo Fisher, Waltham, MA, USA), and the first-strand cDNA was prepared with the PrimeScript RT Kit (Takara, Dalian, China), following the manufacturer’s protocol. The cDNA was used as a template for full-sequence cloning and real-time PCR (qPCR). Use (Takara) thermal cycle program for PCR amplification: 94 °C for 5 min (pre-denaturation), 94 °C for 1 min (deformation), 55 °C for 30 s, 72 °C for 40 s, then 40 cycles, and finally 72 °C for 72 °C 10 min. The amplicon was checked by agarose gel electrophoresis (1%) and recycled with SteadyPure Agarose Gel DNA Purification Kit (Accurate Biotechnology, Changsha, China) to ensure the accuracy of the operation. The relative expression levels of the P450 genes were calculated using the 2^−ΔΔCT^ method [63]. The primers used in this study are shown in Appendix A.

### 5.5. Double-Stranded RNA (dsRNA) Synthesis, RNAi, and Real-Time PCR (qPCR)

Using primers containing T7 RNA polymerase promoter, cytochrome P450 and enhanced green fluorescent protein (EGFP) genes were amplified by PCR. The PCR products were purified by Accurate Biology, and dsRNA was synthesized by using T7 RiboMAX Express RNAi kit (Vazyme, Nanjing, China) according to the manufacturer’s instructions. The concentration of dsRNA was determined by nano-drop spectrophotometer, and its integrity was verified by agarose gel electrophoresis (1.5%).

The RNAi assay was performed by feeding dsRNA to *B. tabaci* MED adults in a feeding chamber (5 cm × 10 cm) according to a previously reported method [64]. After 48 h of feeding on artificial diet with dsRNA, the bioassay was conducted, using four biological replicates per treatment. Mortality was recorded after 24 h of feeding. The differences in the relative expression levels and mortality were assessed following Student’s *t*-test and analysis of variance (ANOVA) using SPSS (v.21) (IBM-SPSS, Armonk, NY, USA), and considered significant at *p* < 0.05.

## Figures and Tables

**Figure 1 ijms-24-05899-f001:**
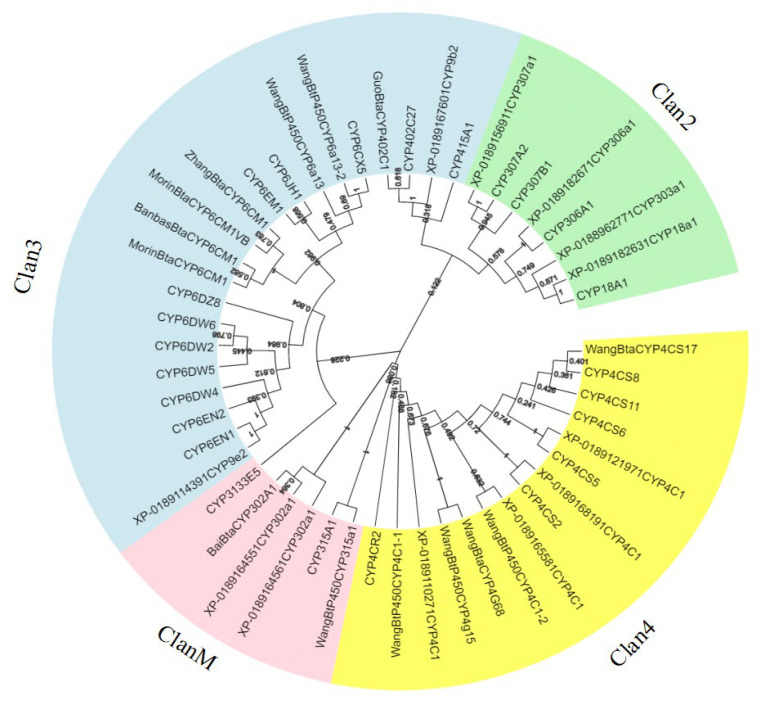
Phylogenetic relationships of P450 families from *Bemisia tabaci* MED. The unrooted phylogentic tree was constructed using MEGA7 by the neighbor-joining method. The bootstrap test was performed with 1000 replicates.

**Figure 2 ijms-24-05899-f002:**
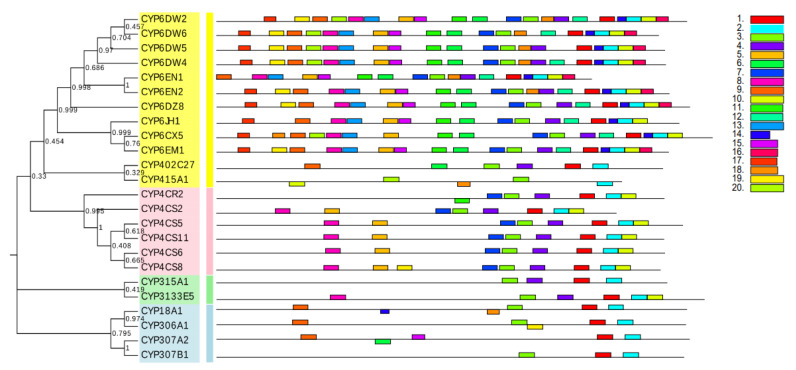
Phylogenetic relationships and protein motif analysis of *Bemisia tabaci* MED CYP. The unrooted phylogenetic tree was constructed using MEGA 7 by the neighbor-joining method and the bootstrap test was performed with 1000 replicates. The colored shadow marks the different CYP families. All motifs were identified by MEME database with the complete amino acid sequences of CYPs. Lengths of motifs for each CYP protein are exhibited proportionally.

**Figure 3 ijms-24-05899-f003:**
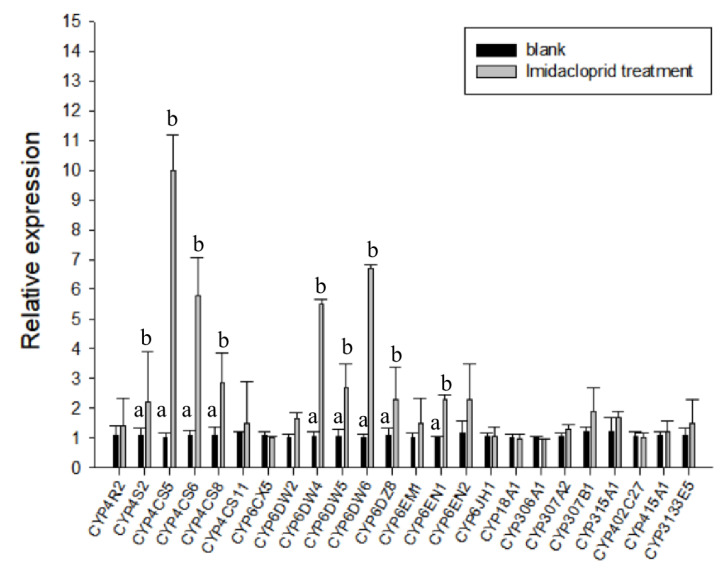
Expression of different genes in *Bemisia tabaci* MED fed imidacloprid (50 ppm). Quantitative real-time PCR (RT-qPCR) analysis of the expression profile of the BtP450 gene superfamily in *B*. *tabaci* MED. Gene names are shown on the *x*-axis and expression levels are shown on the *y*-axis. Different letters on the column indicate that the expression levels of different genes are significantly different (*p <* 0.05).

**Figure 4 ijms-24-05899-f004:**
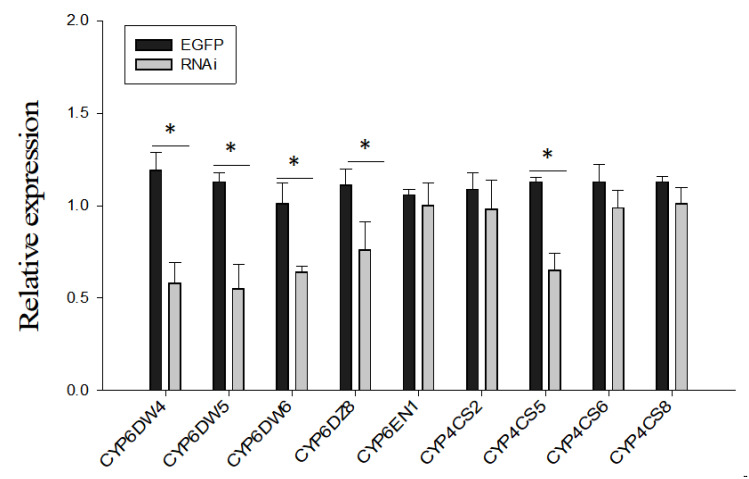
The mRNA level of different P450 in laboratory *Bemisia tabaci* MED after feeding dsRNA. The mRNA levels of P450 were assessed by RT-qPCR after 48 h dsRNA treatment. Values are means ± SE of three biological replicates. The asterisk (*) indicates a significant difference (*p* < 0.05).

**Figure 5 ijms-24-05899-f005:**
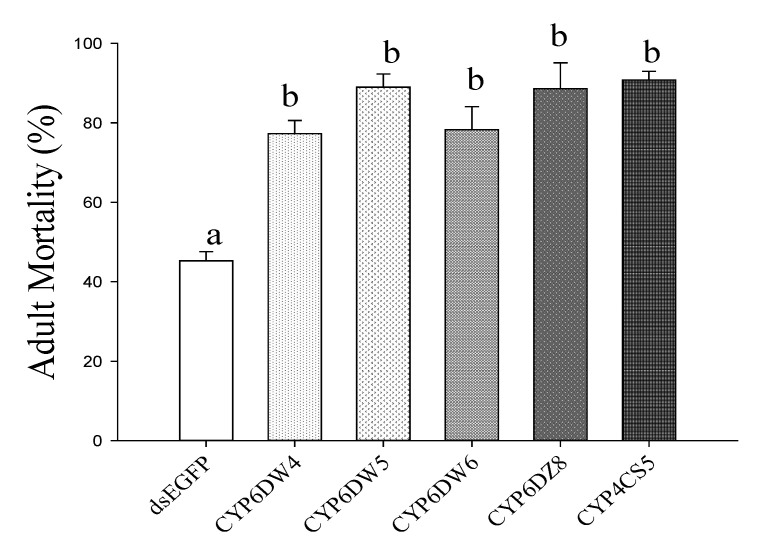
24 h mortality of *Bemisia tabaci* MED fed imidacloprid (100 ppm) after five P450 genes silencing. Means for different letters were significantly different (*p* < 0.05).

**Table 1 ijms-24-05899-t001:** Summary of the *Bemisia tabaci* MED genome assembly.

Data Type	Reads Num	Reads Base	Reads LenN50	Reads LenMean	Reads LenMax
Subreads	7,187,952	105,922,985,976	23,419	14,736	289,016
ZMW reads	6,737,025	100,613,003,720	23,558	14,934	289,016

**Table 2 ijms-24-05899-t002:** Numbers of genes in P450 clans and families identified in *Bemisia tabaci*.

P450 Clan	Clan 2	Clan 3	Clan 4	Clan M
Family	3 (CYP18, CYP306, CYP307)	3 (CYP6, CYP402, CYP415)	1(CYP4)	2 (CYP3133, CYP315)
Subfamilies	4	12	6	2

## Data Availability

All data generated or analysed during this study are included in this published article.

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
