# Peer review of "Genome-Wide Identification and Expression Analysis of the Cytochrome P450 Gene Family in Bemisia tabaci MED and Their Roles in the Insecticide Resistance"

_ijms, 2023, doi:10.3390/ijms24065899_

Round 1
Reviewer 1 Report
The work brings information about a genetic family known to be involved in insecticide resistance. The potential application of the results is relevant. The experimental design and statistical analyzes are conclusive. However, the work has flaws that prevent a complete understanding of the results obtained, as indicated below:
Highlight the importance of the pest in other crops and other geographic locations.
Indicate the dose of use of the insecticide imidacloprid in the field.
Describe PACBIO sequencing methodology.
Specify the breeding time or the estimated number of generations of the insect lineage used.
Remove repeated phrases from lines 155 to 160.
Define Spec activity on line 142.
Explain the relationship of temperature to the stress response to insecticide ingestion on line 111.
The 50 ppm dose in figure 3 is described in the methodology as 5 ppm in line 103, correct with the correct dose.
Explain the difference found from the 24 genes to the 28 genes from the literature in line 87.
Explain the 40% mortality after injection with dsEGFP.
Author Response
Dear Reviewer:
Our manuscript ID (ijms-2232644) has been revised according to the constructive comments of reviewers,the revision explanation as follows:
Point 1: Highlight the importance of the pest in other crops and other geographic locations.
Response 1: Thank you for your suggestion. We added a paragraph to make the content more full. Please look at lines 42-49 of the text.
Point 2: Indicate the dose of use of the insecticide imidacloprid in the field.
Response 2: Thanks for your valuable comment. The resistance of different populations is different, and the LC50 of imidacloprid in this population is 100ppm after preliminary research in the laboratory
Point 3: Describe PACBIO sequencing methodology.
Response 3: Thanks for your valuable comment. PACBIO sequencing method has been added, please see lines 197-201.
Point 4: Specify the breeding time or the estimated number of generations of the insect lineage used.
Response4: Thanks for your valuable comment. We have revised Please see line 192-193
Point 5: Remove repeated phrases from lines 155 to 160.
Response 5: Thanks for your valuable comment. We have corrected.
Point 6: Define Spec activity on line 142.
Response 6: Thanks for your valuable comment. We have modified the sentence to increase the cohesion of the sentence. Please see lines 148-151.
Point 7: Explain the relationship of temperature to the stress response to insecticide ingestion on line 111.
Response 7: Sorry for our negligence. In line 118, change temperature to insecticide.
Point 8: The 50 ppm dose in figure 3 is described in the methodology as 5 ppm in line 103, correct with the correct dose
Response 8: Thanks for your valuable comment. We have corrected.
Point 9: Explain the difference found from the 24 genes to the 28 genes from the literature in line 87.
Response 9: Sorry for our negligence. In line94, 28 has been modified to 24.
Point 10: Explain the 40% mortality after injection with dsEGFP.
Response 10: Thanks for your valuable comment. Because all treatments were treated with imidacloprid, the LC50 of imidacloprid to Bemisia tabaci was about 100ppm, which led to the death rate after injection of dsEGFP
Reviewer 2 Report
The manuscript by Qin et al identified five P450 family genes that are associated with imidacloprid metabolism in Bemisia tabaci MED (Q). The methodology used in gene identification and subsequent bioassays for functional characterization looks good. However, the manuscript would require thorough proofreading. As I went through the manuscript, I highlighted and edited a lot of areas to make it more cohesive. Other than this, the manuscript has good information in it and I recommend it to be accepted pending minor revisions. My comments and suggestions are in the attached file.

Author Response
Dear Reviewer:
Our manuscript ID (ijms-2232644) has been revised according to the constructive comments of reviewers,the revision explanation as follows:
Point 1: Line 14: MED in not sweetpotato whitefly (There is a lot of confusion about B. tabaci naming just remove sweetpotato whitefly, stick to just MED)
Response 1: Thanks for your valuable comment. We have revised it. Please see line 14 and line 42-51
Point 2: Line 64: genetic cluster changes
Response 2: Thanks for your valuable comment. We have revised Please see line 64-76
Point 3: Line 74: Therefore to better understand the resistance mechanism in MED, there is a need to
Response 3: Thanks for your valuable comment. We have revised
Point 4: Line 211: Italicize Bemisia tabaci.
Response 4: Thanks for your valuable comment. We have corrected.
Point 5: Line 217: The MED used in the current study is susceptible or resistant to imidacloprid?
Response 5: Thanks for your valuable comment. As a sensitive population, the population has never been exposed to pesticides since its collection, and it has been raised under laboratory conditions.
Point 6: Line 236: After 48 h of feeding on artificial diet with dsRNA,
Response 6: Thanks for your valuable comment. We have revised, please see line 244-245
Point 7: Line 111: B. tabaci MED insecticide stress response
Response 7: Thanks for your valuable comment. We have revised, please see line 118
Point 8: Line 118: After knocking out these genes, the mortality rate for whiteflies exposed to dsRNA was significantly higher in comparison to whiteflies exposed to dsEGFP
Response 8: Thanks for your valuable comment. We have revised, please see line 125-126
Point 9: Line 131: Previous studies have shown that the P450 gene families: 4, 6, and 9 are associated with pesticide resistance and host adaptation. For example, …
Response 9: Thanks for your valuable comment. We have revised, please see line 137-138.
Point 10: Line 136: The P450 4, 6, and 9 families are Clan 3 and Clan 4 genes in B. tabaci, therefore it is very likely that these clans of genes might be related to the insecticide resistance in B. tabaci as well.
Response 10: Thanks for your valuable comment. We have revised, please see line 142-144
Point 11: Line 139: remove period after enzymes
detoxification enzymes, help ward off and catabolize toxins including pesticides.
Response 11: Thanks for your valuable comment. We have revised, please see line 146
Point 12: Line 141: target genes can lead to an increase in gene expression and diversification of coding sequences with the variable outcome for insecticide metabolism in insects (Li et al., 2007). For instance, in B. tabaci the increase in susceptibility has been primarily associated with the mutations in the P450 monooxygenase rather than the nicotinic acetylcholine receptor, which is the target site of imidacloprid action (Rauch et al., 2003).
Response 12: Thanks for your valuable comment. We have revised, please see line 147-151
Point 13: Line 147: Similarly, P450-mediated effects of insecticide resistance have been reported in aphids
Response 13: Thanks for your valuable comment. We have revised, please see line 1152-153
Point 14: Line 150: Also, Puinean proposed that
Line 159: remove, “Similarly, this phenomenon has also been reported in aphids (Le et al., 160 2006).”
Response 14: Thanks for your valuable comment. Duplicate statement has been deleted.
Point 15: Line 166
Response 15: Thanks for your valuable comment. We have revised, please see line 165-177
Point 16: Line 177 Remove , “Our tissue-specific expression analysis revealed almost ubiquitous expression patterns, indicating some possibly duplicated P450s or unexpressed pseudogenes”
Response 16: Thanks for your valuable comment. We have corrected.